# Keep It Simple: Improving the Ex Situ Culture of *Cystoseira s.l.* to Restore Macroalgal Forests

**DOI:** 10.3390/plants12142615

**Published:** 2023-07-11

**Authors:** Ana Lokovšek, Valentina Pitacco, Domen Trkov, Leon Lojze Zamuda, Annalisa Falace, Martina Orlando-Bonaca

**Affiliations:** 1Marine Biology Station Piran, National Institute of Biology, Fornače 41, 6330 Piran, Slovenia; valentina.pitacco@nib.si (V.P.); domen.trkov@nib.si (D.T.); leon.lojze.zamuda@nib.si (L.L.Z.); martina.orlando@nib.si (M.O.-B.); 2Jožef Stefan International Postgraduate School, Jamova Cesta 39, 1000 Ljubljana, Slovenia; 3Department of Life Science, University of Trieste, Via L. Giorgieri 10, 34127 Trieste, Italy; falace@units.si

**Keywords:** *Gongolaria barbata*, ex situ cultivation, mesocosm, open system, method improvement, brown algal forests restoration

## Abstract

Brown algae from genus *Cystoseira s.l.* form dense underwater forests that represent the most productive areas in the Mediterranean Sea. Due to the combined effects of global and local stressors such as climate change, urbanization, and herbivore outbreaks, there has been a severe decline in brown algal forests in the Mediterranean Sea. Natural recovery of depleted sites is unlikely due to the low dispersal capacity of these species, and efficient techniques to restore such habitats are needed. In this context, the aims of our study were (1) to improve and simplify the current ex situ laboratory protocol for the cultivation of *Gongolaria barbata* by testing the feasibility of some cost-effective and time-efficient techniques on two donor sites of *G. barbata* and (2) to evaluate the survival and growth of young thalli during the laboratory phase and during the most critical five months after out-planting. Specifically, the following ex situ cultivation methods were tested: (A) cultivation on clay tiles in mesocosms with culture water prepared by three different procedures (a) filtered seawater with a 0.22 μm filter membrane, (b) filtered seawater with a 0.7 μm filter membrane (GF), and (c) UV-sterilized water, and (B) cultivation on clay tiles in open laboratory systems. After two weeks, all thalli were fixed to plastic lantern net baskets suspended at a depth of 2 m in the coastal sea (hybrid method), and the algal success was monitored in relation to the different donor sites and cultivation protocol. The satisfactory results of this study indicate that UV-sterilized water is suitable for the cultivation of *G. barbata* in mesocosm, which significantly reduces the cost of the laboratory phase. This opens the possibility of numerous and frequent algal cultures during the reproductive period of the species. Additionally, if the young thalli remain in the lantern net baskets for an extended period of several months, they can grow significantly in the marine environment without being exposed to pressure from herbivorous fish.

## 1. Introduction

Canopy-forming fucoids (Fucales, Phaeophyta) are the dominant habitat builders along the rocky shores of the Mediterranean [1,2]. In particular, *Cystoseira sensu lato* and *Sargassum* species form dense forests that are among the most productive Mediterranean coastal communities [3,4]. These habitats thrive from intertidal to circalittoral bottoms, with diverse species replacing each other along a bathymetric gradient [5]. *Cystoseira sensu lato* species (hereafter referred to as *Cystoseira*) have recently been subdivided into three genera: *Cystoseira*, *Ericaria*, and *Gongolaria* [6]. They are long-lived species [7,8] with relatively large size and high biomass [9]. Important ecosystem services [10] provided by *Cystoseira* algal forests include high primary production [11], rich communities of algae and invertebrates in the understory [12,13,14,15], high fish densities and diversity [16,17], and long-term carbon sink and nutrient cycling [18,19,20]. Other human interests in *Cystoseira* include the production of numerous bioactive metabolites with therapeutic properties [21]. *Cystoseira* spp. species have been declared as “species of Community interest” in the European Habitats Directive (92/43/EEC) [22]. With the exception of *Cystoseira compressa*, all *Cystoseira* spp. are also listed in Annex II (List of Threatened Marine Species of the Mediterranean) of the Barcelona Protocol (UNEP/MAP, 2009) on Specially Protected Areas and Biological Diversity [23]. They are also monitored according to IUCN, RAC/SPA (Regional Activity Centre for Specially Protected Areas, established under the Barcelona Convention), and MedPan (Mediterranean Network of Marine Protected Areas) requirements. Moreover, due to their sensitivity to environmental changes, *Cystoseira* spp. are used as relevant bioindicators for assessing the ecological and environmental status of coastal waters under the Water Framework Directive (WFD, 2000/60/EC) [24] and the Marine Strategy Framework Directive (MSFD, 2008/56/EC) [25].

As a result of anthropogenic stressors, such as coastal urbanization [26,27], marine pollution/eutrophication [28,29,30,31,32,33], and turbidity/sedimentation [29,34], *Cystoseira* populations have gradually disappeared in many coastal areas. They have often been replaced by so-called algal turf [4,27,35,36], sea urchin barrens with coralline algae [37,38], or other upright macroalgae (e.g., Dictyotales and Sphacelaries), where an alternative stable state is established [39]. The low turf mat then prevents the growth of more complex algae [40,41] and even restricts grazing due to the large amount of sediment trapped in the turf.

This process is called biotic homogenization and results in a loss of biodiversity [42]. Human activities in the marine environment also negatively affect connectivity, leading to habitat fragmentation and loss and genetic disjunction even at small spatial scales [34,43]. In addition, the zygotes of *Cystoseira* sink rapidly [7] so that they stick to the substrate near the parent algae. Therefore, due to their low dispersal capacity, colonization of new or damaged areas is difficult, so habitat fragmentation has a major impact on these species. In addition, herbivorous fishes have been shown to be capable of permanently damaging brown algal forests [44] and have been defined as important ecosystem modifiers. In particular, salema (*Sarpa salpa* (L., 1758)), whose abundance has increased, probably due to the overfishing of large piscivorous predators [44,45], is responsible for the lower success of *Cystoseira* restoration efforts in the Adriatic Sea [46,47,48].

Many researchers have also recently reported changes in the distribution and abundance of macroalgal forests due to increasing water temperatures and thermal stresses such as marine heat waves [46,49,50]. Such anomalies can affect the phenology and physiology of canopy-forming species, reduce their performance, increase their susceptibility to other pressures, and lead to population declines and local extinctions [51,52]. Thermal anomalies can also lead to changes in associated species and their interactions [53,54], which can affect ecosystem functions and associated ecosystem services [55,56]. Therefore, the loss of brown algal forests also leads to a reduction in the ability of the oceans to sequester carbon dioxide and contribute to climate change mitigation [57].

The first strategy to address the decline of *Cystoseira* species and their habitat was their protection through international agreements (e.g., Bern Convention [58], Barcelona Convention, Directive 92/43/EEC [23], European Red List of Habitats [59]). Nevertheless, there is little evidence of the natural recovery of degraded *Cystoseira* forests [29,60,61]. For these reasons, restoration actions are needed, also in the context of the UN Decade on Ecosystem Restoration (2021–2030) [62] and the EU Biodiversity Strategy to 2030 [62] (with the proposal of the nature restoration law being its key element). However, implementing successful restoration efforts requires detailed knowledge of the current and past distribution of lost habitats and species, the pressures that have led to their decline, and accurate characterization of donor populations [47,63].

In recent decades, restoration attempts have been made in the Mediterranean to prevent further loss of fucoids, and several methods have been tested, from transplanting adult thalli [64,65,66] or juveniles [29,67], in situ seeding [61,68] and out-planting ex situ grown germlings [10,46,47,48,64,68,69,70,71,72]. Non-destructive in situ and ex situ techniques are best suited for the recovery of endangered species to avoid the depletion of donor populations [10]. In particular, several efforts have been made to develop cost- and time-efficient methods for the ex situ cultivation of *Cystoseira s.l.* species. However, several factors must be considered when cultivating germlings for reforestation, such as water parameters, potential infestation by epiphytic algae, and the development of microbial biofilms on culture surfaces. While epilithic microbial biofilms are an important component of the coastal ecosystem and have numerous beneficial effects, such as serving as primary producers, food resources for herbivorous animals [73,74,75], and promotors of larval settlement and macroalgal recruitment [76], they have also been shown to affect the growth and success of neighboring macroalgae by releasing allelopathic compounds [77] and potentially suppressing their growth.

On the other hand, open laboratory system cultures of *Cystoseira s.l.* can also be negatively affected by grazing pressure, especially at early-life stages [78]. Herbivorous gastropods, crustaceans, and polychaetes can also play a role in consuming various parts of the thalli, potentially affecting its biomass and the overall health of the algae [78]. Grazing plays a crucial role in regulating the density of canopy-forming fucoids and represents a major challenge in marine forest restoration efforts.

The present research focused on the restoration of *G. barbata.* The main objective of the study was to improve and simplify the current ex situ laboratory protocol for the cultivation of *Gongolaria barbata* [46] by testing the feasibility of some cost-effective and time-saving techniques on two populations (donor sites) of *G. barbata* and assessing germling survival and growth during mesocosm culture and during the most critical five months after out-planting (as germlings need to adapt to the coastal sea conditions and start developing articulated thalli). Specifically, the following ex situ cultivation methods were tested: (A) cultivation on clay tiles in mesocosms with culture water prepared with three different procedures and (B) cultivation on clay tiles in open laboratory systems. After the cultivation phase, all thalli were fixed to plastic lantern net baskets suspended at a depth of 2 m in the coastal sea, and the restoration success was monitored in relation to the different donor sites and the cultivation protocol.

## 2. Materials and Methods

### 2.1. Study Area and Material Collection

The study area is located in the Gulf of Trieste, a shallow, semi-enclosed bay in the northernmost part of the Adriatic Sea, with an average depth of only 18.7 m and a maximum depth of 37 m. The area is known for the highest tidal amplitudes (average = 88 cm; [79]) and the lowest temperatures in winter [80]. Water temperature and salinity are strongly influenced by river discharge. The Slovenian part of the Gulf of Trieste accounts for one-third of the total surface area of the Gulf [81]. The coastal relief varies from steep rocky cliffs to gradual sloping beaches. The lower part of the coast, in particular, has been heavily modified by anthropogenic development and urbanization. Today, only one-fifth of the coastline remains in its natural state [79,81]. The bottom along the coast is mainly rocky and consists of alternating layers of flysch, sandstone, and soft marl [79].

A sharp decline in fucoids has been reported in the northern Adriatic Sea [35,36,82], leading to displacement by turf-forming taxa in shallow waters. The increasing abundance of these low-lying algae is likely related to human-induced hydro morphological changes to the coastline and high sediment resuspension rates [82]. In addition, negative impacts by native herbivorous fish have also been documented [46,48]. Currently, *Gongolaria barbata* (Stackhouse) Kuntze and *Cystoseira compressa* (Esper) Gerloff and Nizamuddin are quite common in the Gulf of Trieste on the Slovenian coast, while they have almost disappeared on the Italian coast. Other species of *Cystoseira s.l.* are already rare in Slovenian waters and extinct in the Italian waters of the Gulf of Trieste.

Fertile apices of *G. barbata* were collected from donor populations at two sites:Izola Merkur, which is characterized by healthy and dense populations of *G. barbata* and *C. compressa*; coordinates: 45°32.653, 13°40.554.Izola Belvedere, which is characterized by healthy and dense populations of *G. barbata;* coordinates: 45°31.979, 13°38.488.

The two donor sites (Figure 1) for the collection of apical fronds were selected according to previous studies [48,82] and SCUBA diving surveys in spring 2022.

In early April, the fertile apical fronds of *G. barbata* were collected by SCUBA divers at approximately 2 m depth. We collected up to 10% of fertile apices per individual thalli to avoid a negative effect on the population. The apices were cut with scissors and kept in mesh bags that were stored on the boat in buckets with seawater and immediately transported (within 30 min) to the laboratory of the Marine Biology Station Piran (MBSP).

### 2.2. Laboratory Work

The current protocol for preparing water for the ex situ culture of *G. barbata* [46] requires large volumes of ultra-filtered seawater (0.22 μm filter membrane), which is time-consuming and costly. Therefore, three different water treatments were tested for the present study: (a) filtered seawater with a 0.22 μm filter membrane, (b) filtered seawater with a 0.7 μm filter membrane (GF), and (c) UV-sterilized water.

#### 2.2.1. Preliminary Testing of the Effectiveness of UV Sterilizers

Before using UVC-sterilized water for ex situ cultivation experiments, the efficiency of an Eheim Clear UVC 11 W sterilizer was tested.

Test for bacterial growth on agar plates. Cultures from unsterilized, 1× UVC-sterilized, and 5× UVC-sterilized water (water flowed through sterilizer 5 times at water flow 150 L/h) were inoculated onto growth media plates by (a) adding 100 μL of each water sample to separate agar plates, (b) filtering 50 mL of each water sample through a 0.45 μm filter that was then put on separate agar plates. The agar plates were monitored, and CFU (=colony forming units) were counted daily for 3 days.Test for bacterial and microalgal growth on the clay tiles in the growth chamber. Three 60 cm × 30 cm × 30 cm aquaria were prepared, each containing three clay tiles. Unfiltered seawater (control) was added to the first aquarium, 1× UVC-sterilized water to the second, and 5× UVC-sterilized water to the third. Under established culture conditions (T = 17 °C, photoperiod 15:9 h light:dark cycle, light intensity = 125 μmol photons m^−2^ s^−1^), the clay tiles were monitored daily for 23 days.

#### 2.2.2. Ex situ Cultivation in the Mesocosm

In the laboratory, apices with mature receptacles from two donor populations were cleaned of epibionts with a soft brush, rinsed with filtered seawater, wrapped in aluminum foil, and stored overnight (24 h, approximately 5 °C). This thermal shock triggered the release of gametes from the conceptacles, allowing fertilization [69]. *G. barbata* is a monoecious species; therefore, self-fecundation can occur. In environmentally controlled rooms, the next day, fertile apices from the two populations were placed on clay tiles in aquaria (previously disinfected with 5% bleach solution) with differently treated seawater (Figure 2).

A total of 6 aquaria (40 cm × 34 cm × 9 cm) were prepared, 3 for each donor site; the first with 2× filtered seawater (through GF filter and 0.22 μm pore size filter), the second with only GF filtered seawater, and the third with seawater filtered through a UVC sterilizer (Figure 2). Before the experiment, all aquaria were disinfected with a 5% bleach solution. All aquaria were covered with plexiglass lids to prevent water evaporation. The temperature in the growth chamber was set to 17 °C, and the photoperiod was set to a 15:9 h light:dark cycle. The light intensity was set to approximately 140 μmol photons m^−^^2^ s^−^^1^. For illumination, 4 Osram Fluora Florescent tubes 36 W with a length of 120 cm and a luminous flux of 1400 lumens per container were used.

Twenty-five clay tiles (diameter = 6 cm, with a central hole of 0.6 cm) were placed in each aquarium, and 5–6 apical fronds were carefully placed on each clay tile. On the first day, the aquaria were filled with water to a maximum height of 1 cm (2 mm above the apices) to prevent the apices from floating. Three additional glass slides with apices were added to each aquarium to monitor, measure, and photograph the development of zygotes every 24 h using an Olympus SZX16 stereomicroscope. The apices were left on the tiles for 2 days and then carefully removed with forceps. On day 3, 1 L of treated seawater (GF + 0.22, GF or UVC) was added. From day 6, the water was changed every 2–3 days, and Von Stosch’s solution and germanium dioxide (to prevent diatom contamination) were added [10,69]. On day 6, air pumps and bubblers were also installed in the aquaria. Ten randomly selected tiles from each aquarium were photographed vertically with a Nikon D850 camera after the first and second weeks to assess the coverage of *G. barbata* germlings.

During the first 16 days, the juvenile *G. barbata* thalli were measured and photographed every 24 h under the Olympus SZX16 stereo microscope.

#### 2.2.3. Ex Situ Cultivation in Open System

Two 60-liter aquaria were prepared and supplied with UV-sterilized seawater, one for each donor site. A Kessil H160 Tuna Flora LED grow lamp with a spectrum of 380–780 nm was installed above each aquarium. The photoperiod was set to a 15:9 h light:dark cycle. The light intensity was set to approximately 140 μmol photons m^−^^2^ s^−^^1^. To prevent sediment from settling on the clay tiles and on young algal thalli, the tiles were placed on a net 5 cm above the bottom. A total of 48 clay tiles were placed in each aquarium, and 5–6 apical fronds of *G. barbata* were carefully placed on each clay tile. On the first day, the aquaria were filled with water to a maximum height of 5.5 cm (2 mm above the apices) to prevent the apices from floating. After 48 h, the apical fronds were carefully removed, and aquaria were filled with water to a height of 25 cm. Seawater, pumped directly from the sea in front of the MBSP, flowed directly into an Eheim Clear UVC 11 W sterilizer and then into the aquaria. Sediment accumulating under the net was vacuumed once a week. For the first 16 days, the young *G. barbata* thalli were measured and photographed every 24 h under a stereomicroscope, then once per week with a caliper. After three months, the lighting was changed to the blue spectrum to accelerate the growth of *G. barbata* thalli, and a stream pump (Sicce Voyager nano, Q max 1000 L/h) was added to the aquaria to improve circulation in the system.

### 2.3. Out-Planting in the Marine Environment

After 16 days, the labeled clay tiles from the different experimental conditions in the mesocosm were transferred to lantern net baskets in front of the MBSP (Figure 3). The tiles were secured to the bottom of the lantern net baskets with zip ties. The lantern net baskets were placed at a depth of approximately 2 m and tied to a concrete weight (placed at a depth of 4 m on a sediment bottom), with buoys attached above them. *G. barbata* development was monitored photographically from day 28 by SCUBA diving, initially every two weeks until day 71. From day 71, development was monitored approximately once a month until day 159 (23 weeks). Nutrient levels, water temperature, pH, oxygen saturation, and salinity were measured every 2–3 weeks at a depth of 3 m at the oceanographic buoy Vida (https://www.nib.si/mbp/en/; accessed on 21 February 2023).

### 2.4. Photo-Processing and Statistical Analyses

Photos from the ex situ culture in thermostatic chambers were processed using the ACDSee program and then analyzed using the ImageJ program to calculate the coverage of *G. barbata* germlings on each tile in the first two weeks. Percent cover was determined manually by thresholding and selecting pixels in the color spectrum that corresponded to areas with *G. barbata* germlings [72]. Statistical analyses were performed using R 4.2.2 (R Core Team, 2020). A two-way robust ANOVA based on trimmed means (20% trimming level) [83] was performed for the main effect and interaction to check for differences in thalli length and cover of *G. barbata* germlings between two fixed factors: donor site (with 2 levels: B and M) and treatment (with 4 levels: UV, 0.22, GF, and open system UV). The test was chosen because it is more robust to deviation from normality and homogeneity of variance than the classic two-way ANOVA. A trimmed mean discards a defined percentage at both ends of the distribution, achieving nearly the same amount of power as the mean in the case of a normal distribution and reducing substantially standard error in the presence of outliers [83]. The statistical outcome measure of this test, Q, is interpreted in the same fashion as the traditional F statistic [83]. Tukey’s multiple comparisons of means were used to compare the treatments and the donor sites independently. Analyses were performed using WRS2 packages [83]. Data were then graphically represented with boxplots using the ggplot2 package [84].

For the evaluation of germlings’ length and coverage in the laboratory and thalli length in lantern net baskets, the tiles were considered replicates, but we faced a problem of spatial pseudoreplication, as some of the tiles belonged to the same aquarium. For logistical reasons, it was not possible to have completely independent replicates in the laboratory phase, but careful interspersion and randomization allowed us to reduce the risk of an aquarium effect during the second (in situ) phase. This aspect was taken into account for statistical analyses and in the interpretation of the results, keeping in mind the risk of inflated Type I errors in the case of a simple pseudoreplication [85]. For this reason, two-way ANOVA was not applied to the results of the laboratory phase.

## 3. Results

### 3.1. Preliminary Test of the Efficiency of UV Sterilizers

#### 3.1.1. Evaluation of Bacterial Growth on Agar Plates

Six agar plates were examined on days 2 and 3, and colony-forming units (CFU) were counted on each plate (Table 1). Samples 1, 3, and 5 were prepared by applying 100 μL of each water treatment sample to the culture medium plates (1× UV-sterilized water, 5× UV-sterilized water, and untreated seawater). Samples 2, 4, and 6 were inoculated by filtering 50 mL of each water treatment sample through a 0.45 μm filter using vacuum filtration. No colonies formed in 1× UV-sterilized water on days 2 and 3 (Table 1). The calculated CFU/L showed that 1× UV-sterilized water prevented the growth of bacterial colonies and could be tested for the cultivation of *G. barbata* germlings.

#### 3.1.2. Evaluation of Bacterial and Microalgal Growth on Clay Tiles in the Growth Chamber

The first growth of biofilm on clay tiles in unsterilized seawater was observed on day 15. A brownish biofilm formed on 2 of 3 clay tiles in this aquarium. No growth was detectable on the clay tiles in the aquaria with 1× and 5× UV-sterilized water. On day 18, a brownish biofilm formed on all 3 clay tiles in the aquarium with non-sterilized seawater, while the clay tiles in the sterilized water were still clear. On the 23rd day, the brownish biofilm on the clay tiles in the non-sterilized seawater spread and covered most of the surface of the tiles, while the tiles in the other 2 aquaria still showed no signs of growth. This result proves that the sterilizer works well to prevent the growth of a microbial biofilm, which could suppress the recruitment and growth of *G. barbata* germlings in the first few weeks.

### 3.2. Growth and Survival of Germlings in the First and Second Week in Mesocosm and Open System

On the fourth day, germlings had an average size ranging from 90.41 ± 13.90 μm (Belveder 0.22 μm) to 137.08 ± 45.90 μm (Merkur GF). The first development of diatoms on the tiles in the mesocosms was observed on the sixth day; therefore, GeO_2_ was added to prevent further growth. The average size of the germlings in the mesocosm ranged from 92.09 ± 20.35 μm (Belveder GF) to 119.87 ± 10.10 μm (Merkur UV) and 98.51 ± 12.00 μm in the open systems (Figure 4).

In the first week, the average size of germlings ranged from 156.71 ± 21.09 μm (Belveder open system) to 258.05 ± 20.71 μm (Merkur GF).

In the second week, germlings in the open system were smaller, with the largest individuals up to 400 μm tall, while the longest germlings in the mesocosms were up to 600 μm tall (Figure 4). The average length of the germlings in the open system was 182.02 ± 29.70 μm for the thalli from the Belveder donor site and 299.39 ± 75.05 μm for the thalli from the Merkur donor site.

Coverage of *G. barbata* germlings on clay tiles did not show any clear pattern related to donor sites or water treatments (Figure 5). After 2 weeks, the coverage on most tiles was between 2 and 3% (Figure 5). In the GF treatment, higher coverage was observed on tiles from the Belveder donor site (Belveder GF = 2.96 ± 1.5, Merkur GF = 0.70 ± 0.19). In the treatments UV and 0.22, the opposite pattern was observed, with higher coverage on tiles from Merkur in the second week (Belveder UV = 1.28 ± 1.09, Merkur UV = 1.75 ± 0.85, Belveder 0.22 = 1.67 ± 1.02, Merkur 0.22 = 2.05 ± 0.94). Diatoms were not more abundant in the aquaria with UV-treated seawater than in the other aquaria with filtered water (pers. obs.).

### 3.3. Growth of Germlings in the Lantern Net Baskets

After outplanting the germlings from the mesocosm in the lantern net baskets in the sea (on day 16), their growth accelerated. After two weeks in the lantern net baskets, the germlings quadrupled their average size from 529.14 ± 61.33 μm (about 0.5 mm) to 2291.52 ± 702.30 μm (2.3 mm). After about one month in the lantern net baskets, the average size of the recruits was approx. 6.2 ± 1.9 mm, and after 2 months, they reached an average length of 21 ± 2.6 mm (Figure 6).

At week 15, germlings from the UV-treated open system were the same size as the thalli transferred from the mesocosm to the lantern net baskets (Figure 6; Belveder open system = 25.6 ± 8.9 mm, Merkur open system = 28.9 ± 18.4 mm, Belveder UV = 26.4 ± 9.2 mm, Belveder GF = 23.6 ± 6.1 mm, Belveder 0.22 = 24.7 ± 9 mm, Merkur UV = 24.5 ± 7.8 mm, Merkur GF = 33.2 ± 11.2 mm, Merkur 0.22 = 25.4 ± 12.1 mm). The germlings on the lantern net baskets reached an average length of 32 mm after 122 days (17 weeks), and an average length of 34 mm, and a maximum length of 43 mm after 164 days. There was a difference (2-way ANOVA, *p* < 0.01) only between germlings Belveder GF and Merkur open system and Belveder GF and Merkur 0.22 (Tukey’s test, *p* < 0.01). There was no difference between germlings’ length from UV treatment and the other methods (Tukey’s test, *p* > 0.01).

The young thalli were also exposed to some minor grazing. Some hermit crabs, juvenile fish from the family Blennidae, and a specimen of *Maya crispata* were found in lantern net baskets.

During the experimental period (early March–late September), the temperature was highest in late July (Table 2). Salinity and pH were stable, while O_2_ saturation was highest in June and lowest in late September. In general, nutrient levels were highest in April and May, consistent with the period of accelerated algal growth (see Figure 6, weeks 2–8).

### 3.4. Growth of Germlings in the Open System

After one month in the open system, the biofilm began to grow on the tiles. The seawater temperature was 13 °C at the end of week 4 (early May). The epiphytic brown alga *Ectocarpus siliculosus* (Dillwyn) Lyngbye was detected in the aquaria. The biofilm was removed from the tiles with a soft brush, taking care not to damage the *G. barbata* germlings. To reduce the growth of the biofilm, the lighting was adjusted to a blue spectrum on day 55 (week 7, early June). A decrease in epiphyte and biofilm growth was soon observed, while there was no difference in the *G. barbata* growth. The increased water turbulence after the installation of the stream pump had a positive effect on algal growth, as the thalli became visibly thicker after about 1 week. On day 71 (week 10, mid-June, seawater temperature = 21 °C), some polychaetes (subcl. Errantia, fam. Nerididae, mainly the species *Platynereis dumerilii* (Audouin and Milne Edwards, 1833)) were found in both aquaria (Figure 7). The polychaetes that entered the water through the inflow water pipes grazed on *G. barbata* germlings and incorporated parts of the algae into the outer layer of their tube. Tubes of polychaetes were observed on many clay tiles. Polychaetes were removed manually by brushing each tile and searching for hidden worms in the algal canopies and under the tiles. Almost every week, we also found *Patella* sp. on the tiles. They entered the system in the larval stage and went unnoticed until they attached themselves to the clay tile. After 16 weeks (end of July, seawater temperature = 26 °C), the average length of germlings was 23 ± 5.1 mm for the Belveder and 28 ± 13.6 mm for the Merkur donor population. The maximum length of the Belveder germlings was 48 mm, while the Merkur recruits reached 57 mm. After 23 weeks (middle September, seawater temperature = 24 °C), the average length of the Merkur germlings was 42 ± 9.6 mm, while the Belveder germlings were slightly smaller with an average length of 39 ± 13.8 mm. No significant difference was found between the growth values of the two sites after 23 weeks (2-way ANOVA, *p* > 0.01). The maximum length of Merkur germlings was 62 mm, while the tallest Belveder germlings measured 50 mm.

## 4. Discussion

Restoration is increasingly recognized as an appropriate strategy to actively initiate the recovery of degraded coastal habitats, including brown algal forests. In our study, we tested whether it is possible to grow *G. barbata* with seawater treated only with a UVC sterilizer. This procedure provided promising results, as after 164 days, there was no significant difference in the growth or survival of germlings between UV-treated water and two types of filtered seawater (see Figure 6).

Although the results of the laboratory phase do not provide statistical confidence due to the problem of pseudoreplication, the observational data did not show a pattern suggesting different growth and survival of germlings when UV-sterilized water was used compared to filtered water for cultivation of *G. barbata* in mesocosms. The use of UV sterilizers significantly reduces the cost and labor of the laboratory phase. This opens the possibility of numerous and frequent algal cultures, reducing costs and labor during the reproductive period of *Cystoseira s.l.* species. However, it should be noted that UVC sterilizers do not destroy multicellular organisms or their eggs and larvae and do not prevent outbreaks of epiphytic algae and polychaete invasions in the open system (*pers. obs*.). This was not a problem in the mesocosms because the water was changed every 2–3 days, and therefore, the polychaete larvae and epiphytic algae did not have enough time to settle and develop. Cultivation of *G. barbata* in the open system was rather tricky. The growth of germlings was slower than in the mesocosm (see Figure 4), probably due to the lower temperature of the water (which was pumped directly from the coastal sea), the lack of additional nutrients (Von Stosch medium), and the massive growth of epiphytes (see Figure 7). Switching the lighting to a blue spectrum on day 55 helped reduce the amount of biofilm on the tiles, as this spectrum has been shown to inhibit the growth of certain types of bacteria and cyanobacteria [86,87]. This is due to the fact that blue light has shorter wavelengths (between 400 and 470 nm) and higher energy, which can disrupt cellular processes and metabolic activities, thereby inhibiting their growth [87,88]. The germlings transferred from the UV-treated open system to the lantern net baskets grew rapidly during the first weeks in the sea and were as large as the thalli from the mesocosm by week 15 (late July; see Figure 6). Transferring algal germlings to floating structures reduces contamination with biofilm and filamentous algae that compete with *G. barbata* for available resources and interfere with the growth and development of its germlings [48,71,72], as the constant movement and turbulence of water around floating structures help to disrupt the biofilm and prevent the accumulation of new microorganisms [89].

In addition, lantern net baskets also serve as protection against fish grazers of the species *Sarpa salpa*. Fish perceive floating baskets as unfamiliar objects and do not immediately associate them with food; therefore, germlings in the basket have more chances to survive (*pers. obs*.). A hybrid method involving a carefully controlled laboratory phase and early out-planting to the sea provided satisfactory results in the Slovenian coastal area [47]. The shortened times in the mesocosm resulted in lower culture maintenance and overall costs, which is essential for expanding restoration efforts beyond scientific interest and is now required by international policies and standards [63].

Larval stages of grazing invertebrates were able to enter the open system with pumped seawater, despite UV treatment. The most problematic were the polychaete worms, which appeared in large numbers and were difficult to detect, as they incorporated parts of algae into the outer layer of their tubes. Some tube-building polychaetes have a unique association with algae, which they utilize for tube construction. Tubes are formed by mucus released from specialized glands in polychaete bodies and mixed with sand, sediment particles, or algal parts and filaments to form a structure around the worm’s body [90] and provide additional structural support and camouflage [91]. The polychaete species *P. dumerilii*, identified in the open system, has glandular masses on the parapodia [92]. These glands excrete layers of fine filaments that form an elastic tubular coating into which algae and other particles are incorporated [92]. As the worm grows, it continuously adds new segments to the tube, increasing its length and adding new layers from the inside, making the tube wall thicker [91]. In most cases, when the larva of a tube-dwelling polychaetes settles, it builds the initial tube. The time required for this initial tube formation may vary depending on the species. For some species, the tube can be produced in a matter of minutes by secretions from the body surface, while for others, it may take hours to complete the tube-building process [93,94].

While the removal of *Patella* sp. and other gastropods was quick, since we could easily detach them from the clay tile with a scalpel blade, the removal of polychaetes was more difficult because their tubes were located between germlings, and we had to carefully brush each tile without damaging the thalli. Another disadvantage of the open system was the constant influx of sediment that settled on the algae and the surface of the clay tiles. The amount of sediment and dissolved nutrients increased after stormy and windy weather, and this increased nutrient concentration led to the proliferation of filamentous algae such as *E. siliculosus.* A dieback of *G. barbata* germlings was observed after they were overgrown by these filamentous algae. Epiphytes are known to negatively affect the germlings by limiting light and nutrient availability and, consequently, reducing host fitness and photosynthesis [95,96,97].

Algal growth in the open system and on the lantern net baskets was affected by rising seawater temperatures and nutrient concentrations (see Table 2 and Figure 6). Between the 6th and 8th week of the experiment, when the temperature reached 20 °C (mid–late May), algal growth increased significantly (2-way ANOVA, *p* < 0.01; see Figure 6). Higher temperatures within the species’ optimal range had a positive effect on the growth of *G. barbata*, as the maximum net photosynthesis of the species is between 20 °C and 30 °C [98]. At temperatures above 28 °C, photosynthetic activity begins to decrease, and the synthesis of phenolic compounds such as phlorotannins may be inhibited [99]. Phlorotannins are important anti-grazing, antibacterial, and UV-protecting agents [100,101,102,103]. Nutrients were present in higher concentrations at the beginning of summer and end of summer (see Table 2). The installation of a stream pump (Sicce Voyager nano, Q max 1000 L/h) in the open system at week 13 had an additional positive effect on the growth of *G. barbata* since the thalli became thicker. Exposing young thalli to mild mechanical disturbances such as increased water movement is, therefore, very important to strengthen young *G. barbata* and prepare them for the conditions in the sea. Water movement is vital for nutrient uptake by macroalgae for several reasons. Firstly, it facilitates the distribution of dissolved nutrients, ensuring a continuous supply for algal growth [104]. Secondly, it enhances the mass transfer processes, promoting the exchange of gases and nutrients between the water and algal tissues [105,106]. Lastly, water movement reduces the thickness of the boundary layer (a thin layer of water adjacent to the algal surface that can act as a barrier to nutrient uptake by creating a concentration gradient that hinders nutrient diffusion) and provides mechanical stimulation, improving nutrient contact with algal surfaces and optimizing nutrient uptake rates [107]. These results are also supported by the study of Clausing et al. [72], who found that germlings exposed to mild but repeated stress were more resistant and grew better.

Thalli were kept in lantern net baskets for more than 5 months and were not noticed by the *S. salpa* schools even when they were larger than 5 cm. This result is very important because the main problem for the successful restoration of *G. barbata* populations in the northern Adriatic Sea is the survival of the thalli at the time of their out-planting. *S. sarpa* populations in the Adriatic Sea have increased dramatically over the last decade, probably due to overfishing of large predatory fish. Therefore, the current priority is to develop an efficient method to reduce grazing pressure on *G. barbata* donor and receiving sites. Various fish deterrent devices [44], anti-grazing devices [46], and protective cages [47,48] have been tested, but these methods have many drawbacks (e.g., visual pollution, frequent cleaning of cages to maintain light intensity required for algal growth, damage by fishing lines or net baskets, and swimmers), and are not suitable for use in large areas. Therefore, in parallel with ex situ algal cultivation combined with the hybrid lantern net phase, the development of innovative deterrent devices should be accelerated to reduce this pressure and allow the ecosystem to rebalance. In addition, countermeasures would be useful to regulate herbivore populations, such as motivating fishermen to target species such as *S. salpa. Cystoseira s.l.* species will have no choice but to try to adapt to the multiple impacts of climate change that are already affecting most marine ecosystems [108], but regulation of herbivorous fish populations must be considered an extremely urgent step to be implemented in areas of the Mediterranean where they have been shown to play a major role in the regression of brown algal forests.

## Figures and Tables

**Figure 1 plants-12-02615-f001:**
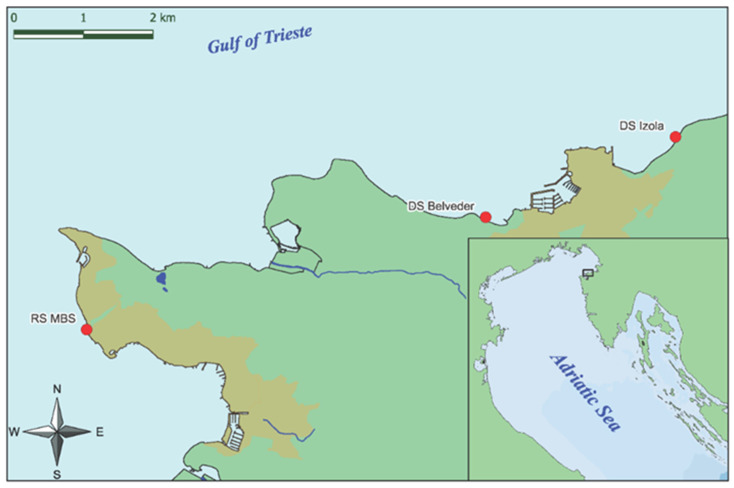
Map of the study area showing donor sites (DS) of fertile apices of *G. barbata* and receiving site (RS).

**Figure 2 plants-12-02615-f002:**
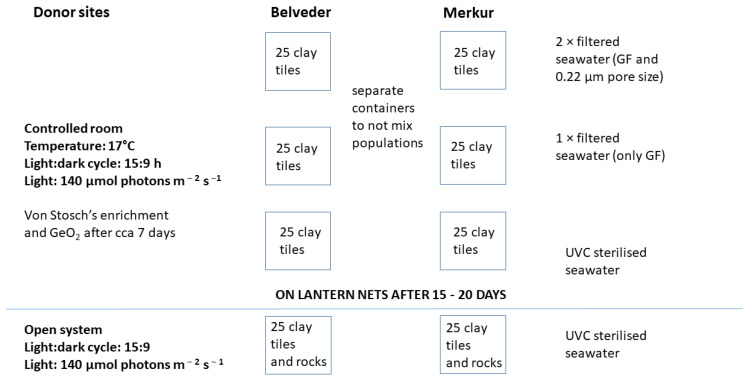
Ex situ cultivation scheme for *G. barbata* at Marine Biology Station Piran in spring 2022.

**Figure 3 plants-12-02615-f003:**
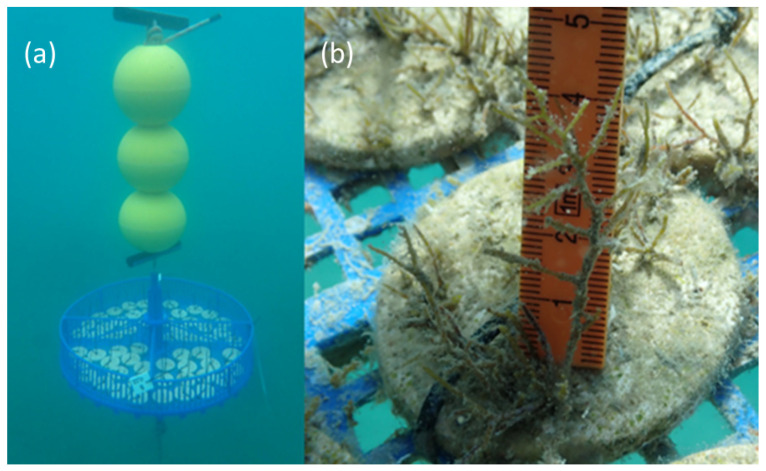
(**a**) A lantern net basket with clay tiles on day 16, and (**b**) measuring the length of *G. barbata* thalli on clay tiles during routine photographic monitoring (M 0.22 on day 122).

**Figure 4 plants-12-02615-f004:**
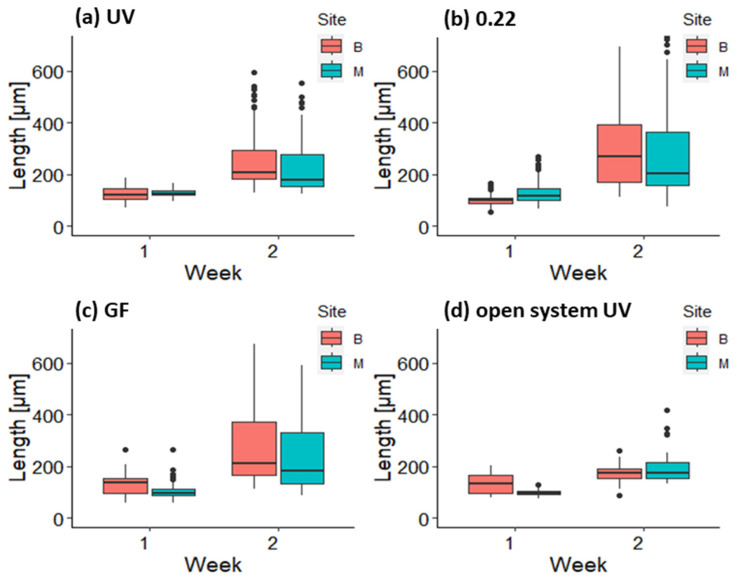
Length (μm) of germlings of *G. barbata* on clay tiles from donor sites B (Belveder) and M (Merkur) during the first two weeks in the three mesocosm treatments: (**a**) UV-sterilized water, (**b**) filtered through 0.22 μm filter, (**c**) filtered through 0.7 μm filter, and (**d**) in the open system.

**Figure 5 plants-12-02615-f005:**
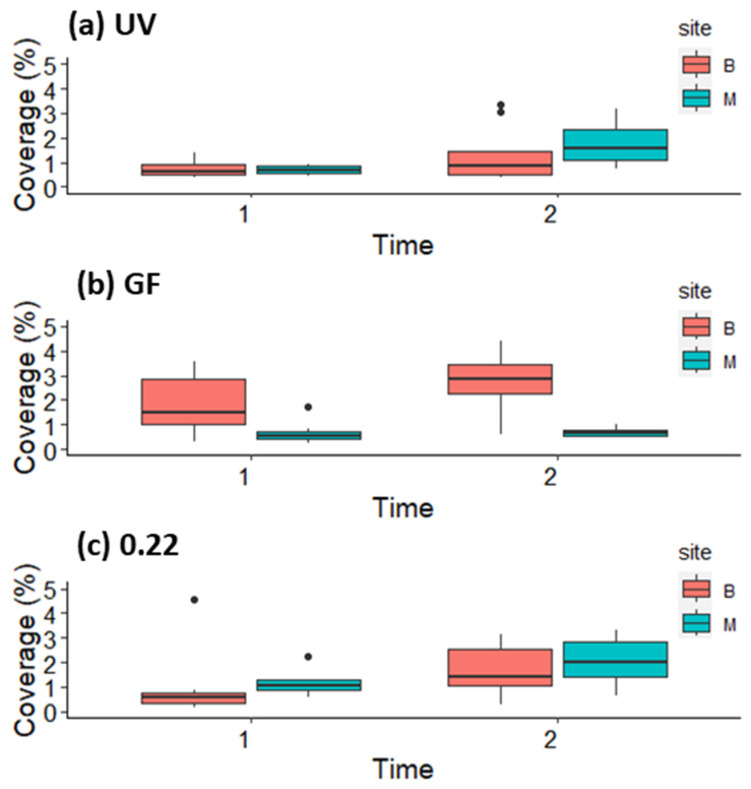
Coverage (%) of *G. barbata* germlings on clay tiles with germlings from donor sites B (Belveder) and M (Merkur) during the first two weeks in the three mesocosm treatments: (**a**) UV-sterilized water, (**b**) filtered through 0.7 μm filter, and (**c**) filtered through 0.22 μm filter.

**Figure 6 plants-12-02615-f006:**
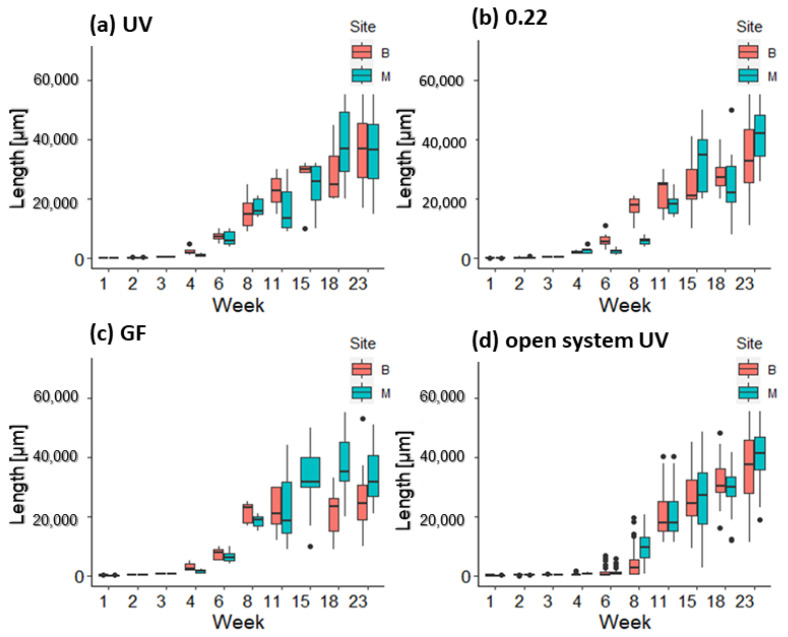
Length (μm) of *G. barbata* germlings on clay tiles from donor sites B (Belveder) and M (Merkur) from different treatments: (**a**) UV = sterilized water, (**b**) 0.22 = water filtered through 0.22 μm filter, (**c**) GF = water filtered through 0.7 μm filter, and (**d**) in the open system, in the laboratory (first 15 days) and on lantern net baskets after out-planting (from day 16).

**Figure 7 plants-12-02615-f007:**
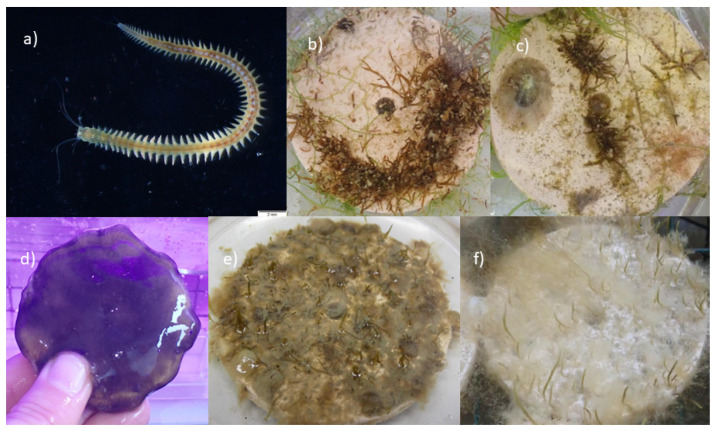
Factors that affected the growth of *G. barbata* thalli in the open system: (**a**) the presence of the polychaete *Platynereis dumerilii*, (**b**) large tube of *P. dumerilii*, which was also made with fragments of *G. barbata*, (**c**) *Patella* sp. grazing on *G. barbata*, (**d**) undetermined gelatinous biofilm, and (**e**,**f**) *Ectocarpus* spp. and undetermined biofilm growing on clay tiles.

**Table 1 plants-12-02615-t001:** Results of tests with differently treated seawater on agar plates (SW—untreated seawater, TNTC—too numerous to count, CFU—number of colony forming units).

Day	Sample_ID	Sample Treatment	Counting Method	Volume of Seawater [μL]	CFU	CFU/L
2	1	SW	On agar plate	100	22	220,000
2	2	SW	On filter	10,000	>250	TNTC
2	3	1× UVC	On agar plate	100	0	0
2	4	1× UVC	On filter	50,000	11	220
2	5	5× UVC	On agar plate	100	0	0
2	6	5× UVC	On filter	50,000	1	20
3	1	SW	On agar plate	100	26	260,000
3	2	SW	On filter	10,000	>250	TNTC
3	3	1× UVC	On agar plate	100	0	0
3	4	1× UVC	On filter	50,000	12	240
3	5	5× UV	On agar plate	100	1	10,000
3	6	5× UV	On filter	50,000	2	40

**Table 2 plants-12-02615-t002:** Seawater parameters in Slovenian coastal waters from March to September 2022 (salinity, pH, O_2_ saturation, and nutrients). Data were collected at a depth of 3 m at the oceanographic buoy Vida (https://www.nib.si/mbp/en/, accessed on 6 July 2023).

Date and Time	Depth [m]	T [°C]	Salinity	pH	O_2_ Saturation [%]	PO_4_ [mg/L]	NO_2_ [mg/L]	NO_3_ [mg/L]	NH_4_ [mg/L]	SiO_3_ [mg/L]
15 March 2022 09:40	3	10.31	38.86	8.18	102.6	0.01	0.098	0.12	0.03	2.06
4 April 2022 09:00	3	11.09	38.46	8.21	100.89	0.01	0.006	0.36	0.03	2.68
12 April 2022 10:20	3	12.05	38.31	8.19	103.08	0.01	0.028	0.72	0.03	2.85
26 April 2022 08:50	3	13.02	38.46	8.19	104.78	0.02	0.006	0.03	0.03	1.8
17 May 2022 09:10	3	20.61	38.36	8.16	108.97	0.02	0.006	0.03	0.03	2.07
31 May 2022 09:50	3	19.3	38.47	8.16	104.37	0.03	0.006	0.08	0.03	1.37
14 June 2022 13:40	3	21.43	38.24	8.15	109.91	0.01	0.014	0.12	0.03	0.56
28 June 2022 08:45	3	25.96	38.09	8.14	104.41	0.01	0.016	0.17	0.03	0.78
12 July 2022 09:50	3	24.64	38.16	8.02	104.42	0.01	0.01	0.18	0.04	1.11
27 July 2022 11:40	3	26.22	38.1	8.15	104.01	0.01	0.006	0.07	0.03	0.5
17 August 2022 09:30	3	24.97	38.42	8.15	102.35	0.01	0.006	0.11	0.03	0.94
30 August 2022 10:00	3	25.22	38.42	8.15	102.16	0.01	0.006	0.18	0.21	0.84
13 September 2022 09:13	3	24.79	38.36	8.15	99.9	0.01	0.022	0.26	0.67	1.08
27 September 2022 10:43	3	21.78	38.27	8.12	95.59	0.01	0.07	0.55	0.84	3.48

## Data Availability

The data presented in this study are available on request from the corresponding author. The data are not publicly available since they originate from the research program funded by the Slovenian Research Agency (ARRS).

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
