# Peer review of "Keep It Simple: Improving the Ex Situ Culture of Cystoseira s.l. to Restore Macroalgal Forests"

_plants, 2023, doi:10.3390/plants12142615_

Round 1

Reviewer 1 Report

The study objectives are firstly to improve and simplify the current ex-situ laboratory protocol for the cultivation of Gongolaria barbata by raising seedlings from two donor sites in three differently filtered types of water and one open system. Due to a lack of replication in the laboratory phase, this objective cannot be said to have been met with statistical certainty, but by observational data only. Any statistical testing done in this phase should be excluded. However, the actual measurements of germling growth and percent coverage in the non-replicated different treatments as well as observations on occurring epiphytes and epifauna are of scientific interest and should perhaps be more emphasised in introduction, which is now only in a very general focus of negative human effects on habitat. A few paragraphs of more details relevant to the study, such as the effects of bacteria or biofilm, expected germling growth/survival and if previously known, the impact of tube-forming worms on germling density, would make the introduction more relevant and provide the reader with useful background to interpret the results.

A second objective was to monitor growth and survival (site and treatment) in situ. Here, each tile functions as a true replicate and this is the part of the study where statistics can be applied. This objective is met.

In general, I do not see the point of a layout putting methods after results and discussion. It confuses the reader since a lot of the information in methods is needed to understand results. The effect is that I have to scroll down after reading introduction because I cannot evaluate the results without understanding the method. Is this a format wished by the authors or the journal? Or just a digital hiccup? Whoever is responsible, please consider the “classic” layout. It makes for easier reading and understanding, especially of a paper like this. I can see that part of the numbering in method is meant to be in the classic form. Whatever is chosen, references to figures and sub section numbers must be corrected throughout the text.

42 double blank space

Table 1. Text is missing explanation of CFU abbreviation

139, 142 and 150. Any difference in this context is, per definition statistically significant, or there is not a difference. Writing statistically significant difference is like a double dip, salsa sauce for example. Since the test and p-value is added after, it is thus inferred that the difference has been statistically tested and proven, thus making writing “statistically significant” superfluous.

167. Suggest adding “in the three mesocosm treatments” to clarify in the figure legend.

171-172. Here you change phrasing from germling to seedling. Why? Either explain (after size so and so, they are called seedlings) or stick to the same word throughout. Go through the whole text as it pops up here and there.

121-129. 2.1.2  The effects of biofilm on settlement should be better presented. As a reader I am lacking information as to whether or not a biofilm is considered positive or negative. I suggest adding a paragraph in the introduction on the effect of biofilm on settling zygotes, as this is a big part of the “why” of the study. For some species, a biofilm might assist in settling. Bacteria and microalgae biofilm are not the same as algal turf, which is mentioned.

140 remove statistical analysis as no replicates were made.

143 remove statistical analysis here, too. A comparison will do. It is still interesting results.

150 and 152 remove statistical analysis here, too. Pers. obs is much more valuable than shaky statistics.

163 Figure 1. Incomplete information in legend. Add unit (μm) after Length and explanation to GF abbreviation, and  after 1 and 2 weeks post settling. General comment for all figure and table texts is to go through them and add enough so that they can be read independently. See journals Advice to author- guide.

167 Figure 2. Incomplete information in legend. Add Percentage coverage of.. or Coverage (%) of…and that it compares two different sites; Belveder(B) and Merkur (M) and in three different treatments a)UV-sterilized water, b) whatever GF means and c)filtered through 0.22 micron filter. By naming the charts with letters instead of treatment, the same text explaining each chart can be used for all similar charts, as this should be in the legend for each one.

171 Reminder, use lantern net baskets or similar wording, as the image of something growing “on nets” is quite different to something growing “in baskets”.

176 I will assume Figure 6 the one now marked as Figure 3 just below. What has happened with the format? Go through all text thoroughly so that the numbering is correct.

177 From the placing out into the sea, statistical comparisons can be made.

187-188 …were found in the lantern net baskets.

193. Figure 3. Figure legend needs to be more explanatory, see previous comment on this. Like adding at which week they were placed in the field, again the name of the sites, the abbreviations if marking them a-d is not used, or the explanations if it is used.

198 Table 2.

204 suggest more clarifying title Growth of germlings remaining in the open system

212 “had a positive effect on G. barbata growth”, since filamentous algae was just mentioned, to clarify it was not Ectocarpus that grew better.

220. How did Patella enter the system?

232 Figure 4. Image A does not really add anything to this context. I suggest removing it, or replacing it with a close up image of germlings, perhaps with some grazing damage if you have one. Just to give an idea of what they look like.. You must name, at least to some taxonomic level, the species of algae shown in F, G and H. This is, after all, a botanical journal. If not, remove these pictures.

235 epiphytic algae, not epiphitic

242 You cannot state this, see comments on replication and statistics. It is not until they are out in the sea that you can truly say that you compare growth between treatments. Thus, Figure 1 and 2 are data still in the lab, and thus not statistically testable, and thus you cannot use the word significant.

250. But you did have to remove them manually, so it added work, did it not? Would be helpful with a mention of the polychaete species use of the germlings for making tubes. Not all polychaetes make tubes. Explain so that the reader understands HOW the polychaete affects/uses the algae, here or in discussion.

252. Incorrect use of “therefore”. Also, you are jumping between tenses here. When you talk about what you did, it is often easiest to use past tense: in this study was tricky. But it means you have to alter the sentences above as well, so on 250: This did not represent a problem in the mesocosms as the water was changed every 2-3 days and therefore the polychaete larvae and epiphytic algae did not have enough time to develop. Go through this throughout the text.

253 probably due to

256 system, not sistem

258 How does the floating structure reduce biofilm? Is it deeper down so less light, or what is the actual cause? Develop some reasoning around this.

265 it sounds like they were intentionally introduced. I assume this was not the case. Formulate so that it becomes clear that despite water filtration, larval stages were able to get into the system from the seawater.

273 Is this increase in nutrients the same as the one shown in table 2? If so, clarify by stating which nutrients. You do not mention measuring nutrients in the open system.

279 not “on nets” see previous comments

288 mild mechanical disturbances, perhaps. Specify what type of disturbance. Might it not also be that increase in water motion reduces the boundary layer and thus makes carbon and nutrient uptake more efficient?

293 not higher, taller.

345 kept in mesh bags, or placed into mesh bags.

364 This is only testing one replicate, as all tiles of one treatment were in the same tank. This is addressed in lines 449-455, thus it would be more helpful to have methods before results. I am not very well versed in statistics but do know that when you have all eggs in one basket, as it were, the factor aquarium can be very tricky indeed. I am pleased to see that you do acknowledge it and take it into consideration in your interpretation. However, there is nothing to say that it is the water sterilization that is the factor tested and not the effect of tank. I suggest you present results from the biofilm experiment without statistical analysis, as no calculations can possibly get around the lack of replication. It will still be interesting and valuable information, if presented well in a clear context.

373 This thermal shock triggers the release of gametes from the conceptacles, allowing fertilization. For those not familiar with this species, I suggest you also mention that it is a monoiceous species, either here or in the paragraph on sampling in the field, as some members of Fucales are dioecious.

380-401. Again, there is no true replication here, as there is only one tank per treatment per site. See previous comment on this. Here, at least, there are 2 tanks per water treatment, so the effect of this might be discussed here. Since all tiles in one treatment are in one tank, it is not until they are planted out in the marine environment that you can begin collect data that can be statistically analyzed. All data prior to this should only be presented as it is, which you have done very nicely in Figure 3.

391 –393 This sentence needs to be re-written to make sense. Do you mean that three apices per aquarium were (randomly?) selected and taken out of the tank every 24 hours for measurements/photography? Or were there 3 apices per tank that were placed on a glass slide, also placed in the tank, and these glass slides with apices were lifted out and documented every 24 hours? Or were they even in the tank? Some clarification and language check is necessary here in order to get it right.

402 2.2.3 Here it would be relevant to add the water flow rate, as you mention in the mesocosm setup that you took care to not have the apices floating about. Did they float about, thus making your careful placement of them unnecessary, or did they stay put? What water depth did you keep in the tanks? You mention it in the mesocosm experiment but not here. Please add this.

416-417 It would be of interest to add the increased flow rate if possible. May I suggest the term circulation, as in “…added to improve the circulation in the system.”

420 Figure 7, if “method placed last”-layout is chosen

427 Figure 7. This is not what I would call a lantern net. Use the term lantern net basket throughout instead. Before I saw the picture, I thought they were grown out in actual lantern nets, thus excluding most fish, but also adding some shade. From the picture, no net was added to the setup. Or was it? If so, use the term lantern nets. If not; lantern net baskets. Apply to whole text.

449-455 I agree that as long as you randomized the placement of tiles in the lantern net tray, any data on growth post field deployment can be used. But not before, when they are all in the same tank. I have done similar experiments myself and gotten round the problem of “factor tank” by using transparent plastic jars with transparent lids, one tile in each jar. Makes for more work but holds up for statistical analysis.

See comments in Suggestions. Minor spelling errors, some inconsistency in choice of word (germling/seedling) and adjust "lantern net" to "lantern net baskets" to not mislead

Author Response

Dear reviewer 1, please find detailed answers to your comments in the attachment. 

Reviewer 2 Report

Please revise the manuscripts by referring to corrections and supplements

Because there are overlapping words and incorrent expressions, it is necessary to review by native speakers

Author Response

Dear reviewer 2, 
Please find answers to your comments in the attachment. 

Round 2

Reviewer 1 Report

Dear Authors.

Well done on improving the paper. It now reads clear and logically throughout and is a very interesting study for all of us working with seaweed and marine restoration. The discussion, in particular, has really been much improved and sets your results in a much clearer context now.

Just a few small suggestions to consider.

Figure 4 legend: Add the information that is in Figure 5 below here as well, regarding two donor sites, etc. All other legends are ok, but you might have missed this one.

I suggest that you name all charts in figures a), b), c) etc and then describe them in the legend. It makes it easier for the reader, as there is less info in the actual chart.

Figure 5 Legend. You have done as I have suggested above in the text, but not altered the figures numbering/lettering. Remember to do so.

Figure 7 legend: Factors might be a better term than Pressures.

For D, add "gelatinous" to biofilm, so that the reader understands even if the image might be printed in poor resolution. It is most likely a cyanobacteria. It looks familiar.

Also, stick to the same format throughout. You did A) in Figure 4, here A- .

Choose one. I prefer the first :-)

Row 916 You have a double "because" in the sentence. Change the first one to "since", or the second one to "as". Or alter both.

Author Response

Dear reviewer 1, please see the attachment with replies to your comments. 
